# Function of Anthocyanin and Chlorophyll Metabolic Pathways in the Floral Sepals Color Formation in Different Hydrangea Cultivars

**DOI:** 10.3390/plants14050742

**Published:** 2025-02-28

**Authors:** Yanguo Ke, Umair Ashraf, Dongdong Wang, Waseem Hassan, Ying Zou, Ying Qi, Yiwei Zhou, Farhat Abbas

**Affiliations:** 1Yunnan Urban Agricultural Engineering & Technological Research Center, College of Economics and Management, Kunming University, Kunming 650208, China; keyanguo@kmu.edu.cn; 2Department of Botany, Division of Science and Technology, University of Education, Lahore 54770, Pakistan; umair.ashraf@ue.edu.pk; 3Key Laboratory of Biology and Genetic Improvement of Horticultural Crops-South China/Guangdong Litchi Engineering Research Center, College of Horticulture, South China Agricultural University, Guangzhou 510642, China; wangdd6768@163.com; 4Department of Soil and Environmental Sciences, Muhammad Nawaz Shareef University of Agriculture, Multan 60001, Pakistan; waseem.hassan@mnsuam.edu.pk; 5College of Agronomy, Yunnan Key Laboratory of Konjac Biology, Yunnan Urban Agricultural Engineering and Technological Research Center, Kunming University, Kunming 650214, China; 2309510041@kmu.edu.cn (Y.Z.); qiying@kmu.edu.cn (Y.Q.); 6Guangdong Provincial Key Laboratory of Ornamental Plant Germplasm Innovation and Utilization, Environmental Horticulture Research Institute, Guangdong Academy of Agricultural Sciences, Guangzhou 510640, China; 7Institute of Tropical Fruit Trees, Hainan Academy of Agricultural Sciences/Key Laboratory of Genetic Resources Evaluation and Utilization of Tropical Fruits and Vegetables (Co-Construction by Ministry and Province), Ministry of Agriculture and Rural Affairs, Haikou 571100, China; 8Key Laboratory of Tropical Fruit Tree Biology of Hainan Province, Haikou 571100, China

**Keywords:** *Hydrangea macrophylla*, floral color, anthocyanin biosynthesis, metabolomics, transcriptome

## Abstract

Hydrangea (*Hydrangea macrophylla*) is distinguished by having sepals instead of real petals, a trait that facilitates color diversity. Floral color is largely predetermined by structural genes linked to anthocyanin production, but the genetic factors determining floral hue in this non-model plant remain unclear. Anthocyanin metabolites, transcriptome, and the CIE*L*a*b** hue system were employed to elucidate the biochemical and molecular mechanisms of floral color formation in three hydrangea cultivars: ‘DB’ (deep blue), ‘LB’ (light blue), and ‘GB’ (green blue). UPLC-MS/MS identified 47 metabolites, with delphinidin, cyanidin, malvidin, petunidin, pelargonidin, and peonidin being prominent. Delphinidins were 90% of the primary component in ‘DB’. The dataset identifies 51 and 31 DEGs associated with anthocyanin, flavonoid, and chlorophyll biosynthesis, with *CHS*, *CHI*, *F3H*, *F3′5′H*, *DFR*, *ANS*, *BZ1*, and *3AT* displaying the highest expression in ‘DB’. Notably, *DFR* (cluster-46471.3) exhibits high expression in ‘DB’ while being down-regulated in ‘LB’ and ‘GB’, correlating with higher anthocyanin levels in floral pigmentation. Comparative analyses of ‘LB’ vs. ‘DB’, ‘DB’ vs. ‘GB’, and ‘LB’ vs. ‘GB’ revealed 460, 490, and 444 differentially expressed TFs, respectively. WRKY, ERF, bHLH, NAC, and AP2/ERF showed the highest expression in ‘DB’, aligning with the color formation and key anthocyanin biosynthesis-related gene expression. The findings reveal the molecular mechanisms behind floral pigmentation variations and lay the groundwork for future hydrangea breeding programs.

## 1. Introduction

*Hydrangea* (Linn.), commonly called ‘Ba Xian Hua’, is a genus within the Hydrangeae tribe, part of the Hydrangeaceae family. Hydrangeas come in 73 different species, 46 of which are native to China [1,2]. China is regarded as the homeland and distribution center of the hydrangea. Members of the genus *Hydrangea* are often categorized into five sections, with bigleaf hydrangea (*H. macrophylla*) being the most widely cultivated hydrangea species, originating from southern China and Japan [3,4].

*H. macrophylla* has a substantial inflorescence and vibrant hues, making it one of the most favored ornamental flowers globally. The species is cultivated as a cut flower and flowering potted plant, esteemed for its enormous and vividly colored inflorescences [5]. The original hue of the flower, namely the sepal, is blue. Nonetheless, diverse breeding initiatives have produced an extensive spectrum of colors, from blue to pink and purple to red, consistently or contingent upon the soil’s cultivation circumstances [6,7,8]. It is well-established that the sepal typically appears blue in acidic soil, while it assumes a crimson hue in alkaline soil. Under acidic conditions, the aluminum ion (Al^3+^) exhibits significant water solubility, facilitating its absorption and transit to the sepal, where it chelates anthocyanins [9,10]. Consequently, producers and researchers have consistently pursued *H. macrophylla* breeding in various colors to improve its aesthetic appeal.

Floral color is a crucial characteristic of ornamental plants, directly associated with the concentration and distribution of pigments [11]. Multiple studies have demonstrated that carotenoids, flavonoids, and chlorophylls are significant contributors to influencing the coloration of flowers [12]. Carotenoids contribute to flowers’ diverse coloration, producing hues ranging from vivid red to orange and yellow. Conversely, flavonoids, recognized as the predominant secondary metabolites in plants, generate hues that vary from light yellow to purple, depending upon the specific type of flavonoids present [13,14]. A single pigment, a mixture of several pigments (e.g., chlorophylls, anthocyanins, and carotenoids), or a lack of pigments altogether leads to the colors seen in the petals and sepals. The blue coloration of hydrangea sepals has been studied since the early 20th century. The main anthocyanin is 3-*O*-glucosyl delphinidin, with co-pigments including neochlorogenic acid, 5-*O*-p-coumaroylquinic acid, chlorogenic acid, and Al^3+^ [7,15,16].

Anthocyanins, classified as flavonoids, are significant secondary metabolites produced by plants. These widely distributed, water-soluble compounds assist several ornamental plants in developing their colors. Plants have six main anthocyanidins based on benzene ring substituents: delphinidin, petunidin, cyanidin, malvidin, pelargonidin, and peonidin [17,18]. Anthocyanins contribute to the orange, red, magenta, violet, and blue colors of flowers and enhance plants’ ability to adapt to various environmental stresses, including pathogen infection, UV radiation, drought, and low temperatures, therefore improving their aesthetic appeal [19]. Many enzymes and transcription factors regulate the intricate mechanism of anthocyanin production and accumulation, which is subsequently driven by a wide range of environmental variables, including light, water stress, and temperature [20,21]. *Hydrangea* cultivars show varying colors even in the same soil, highlighting intricate gene regulatory mechanisms and the association between secondary metabolites. Most studies on hydrangeas have focused on their blue pigmentation, leaving the molecular regulation of anthocyanin synthesis for *H*. *macrophylla* sepal color untouched. Consequently, further investigation is necessary to understand the mechanism governing sepal pigmentation in *H. macrophylla*.

Formerly, much emphasis has been placed on the influence of Al^3+^ and acidic soil on the process of hydrangea flower coloration; nevertheless, the molecular mechanisms and critical compounds that contribute to color development have received limited investigation. Therefore, the current research was performed to elucidate the variations in anthocyanin derivatives and the underlying molecular mechanisms in three distinct colored sepal tissues of *H. macrophylla* cultivars during full bloom stages. Differentially accumulation metabolites (DAMs) and differentially expressed genes (DEGs) were identified and analyzed. This approach is aimed at clarifying the essential genes responsible for flower color variation in hydrangeas and examining the prevalent or distinct mechanisms that contribute to this variation, ultimately providing new insights for breeding cultivars with specific sepal colors, particularly those exhibiting blends of blue, green, and multiple colors.

## 2. Results

### 2.1. Phenotypic Characterization of Hydrangea Cultivars

The coloration of flowers typically results from the large amount of carotenoids, flavonoids, and chlorophylls. Three representative-colored cultivars were selected for this study to explore the production of pigments in hydrangea. The visual examination of the three hydrangea cultivars revealed that the sepals of ‘Bo Da Lan’, ‘Qian Lan’, and ‘Lv Lan’ exhibited deep blue (‘DB’), light blue (‘LB’), and green with blue edges (‘GB’) colors, respectively (Figure 1A).

To assess the sepal color, we employed the CIELAB technique to specify multiple leaf color indices (*L**, *a**, *b**) and analyze pigment concentrations. The *L** (lightness) parameter ranges from 100 (white) to 0 (black); the positive *a** value signifies a predominance of red over green, while the positive *b** value suggests a predominance of yellow over blue. The *L*a*b** three-dimensional coordinates indicated that the *L**, *a**, and *b** values among the three cultivars were markedly distinct (Figure 1B–D). The ‘DB’ exhibits the highest value of *a** and the lowest value of *b**, whereas the ‘GB’ displays the highest value of *b** and the lowest value of *a**. The maximum *L** value was observed in the ‘LB’ cultivar (Figure 1D).

We measured the chlorophyll pigment concentrations in the sepals of the three samples. Significant disparities in photosynthetic pigments were observed among the three samples. The cultivar ‘GB’ exhibited superior chlorophyll content (SPAD values) relative to the ‘DB’ and ‘LB’. The chlorophyll content was approximately 2- to 3-fold higher than the other two cultivars. Likewise, the cultivar ’DB’ exhibits significantly higher chlorophyll content than the cultivar ‘LB’ (Figure 1E).

### 2.2. Metabolomic Profiling of Hydrangea Cultivars

To determine the role of anthocyanin-related metabolites in the floral arrangements of hydrangea cultivars, extracts from the sepals of three hydrangea varieties were analyzed using Ultra Performance Liquid Chromatography Tandem Mass Spectrometry (UPLC-MS/MS). To gain insights into the broader attributes of the metabolite dataset, we initially conducted principal component analysis (PCA) (Figure 2A). Principal Component 1 and Principal Component 2 accounted for 61.72% and 34.76% of the metabolite distribution in the samples, respectively. The three sample groups were differentiated, demonstrating notable variations in anthocyanin contents across different cultivars.

The anthocyanins extracted from ‘LB,’ ‘DB,’ and ‘GB’ exhibited similar spectral properties to anthocyanin standards. By cross-referencing the mass spectrometry data from ‘DB’ samples with compound information in the database, we identified 47 anthocyanin-related metabolites. This includes 11 cyanidins, 6 delphinidins, 5 malvidins, 4 pelargonidins, 5 peonidins, 4 petunidins, 3 procyanidins, and 8 flavonoids. Of the 47 anthocyanin metabolites analyzed, 25 were identified across all cultivars (Figure 2B). The cultivars ‘GB’, ‘LB’, and ‘DB’ were associated with 4, 1, and 2 specific compounds, respectively. The total anthocyanin content enhanced progressively from ‘GB’ to ‘LB’ and then to ‘DB’, measuring 38.85, 608.89, and 1517.87 ug/g, respectively (Figure 2C; Appendix A). Flavonoids constituted the primary components of ‘GB’, comprising 89% of the total content. In the ‘LB’ cultivar, delphinidins and flavonoids constituted the primary pigment components, accounting for 62% and 35%, respectively. Delphinidins constituted 90% of the dominant composition of ’DB’. The delphinidin concentrations were 7%, 35%, and 90% in the samples labeled ‘GB’, ‘LB’, and ‘DB’, respectively. The significant presence of delphinidin-3-*O*-glucoside is likely the primary factor contributing to the deeper blue hue observed in the sepals of ‘DB’ compared to ‘LB’ and ‘GB’.

Clustering by hierarchy heat map assessment of 35 differential accumulation metabolites (DAMs) revealed that certain major pigment substances with comparatively high concentrations varied with cultivar changes (Figure 2D). The data shows that cyanidin-3-*O*-galactoside, malvidin-3-*O*-arabinoside, delphinidin-3-*O*-(6-*O*-p-coumaroyl)-glucoside, peonidin-3-*O*-sambubioside-5-glucoside, malvidin-3-*O*-galactoside, and pelargonidin-3-*O*-galactoside were among the main compounds in ‘GB’. In ‘DB’ and ‘LB’, cyanidin, delphinidin, pelargonidin, petunidin, and peonidin-3-*O*-(6-*O*-p-coumaroyl)-glucoside were the major metabolites.

K-means clustering categorized the anthocyanin content across three cultivars into 10 distinct clusters (Figure 2E). Cluster 5 comprised 8 metabolites, cluster 6 contained 7, clusters 1 and 8 each had 4, clusters 3 and 4 featured 3, and clusters 2 and 10 possessed 2 metabolites. Meanwhile, clusters 7 and 9 comprise single anthocyanin metabolites, exhibiting distinct expression profiles across three cultivars. This signifies that a specific anthocyanin serves an essential function in sepal color development in certain hydrangea cultivars.

### 2.3. Analysis of the Transcriptome in the Sepals of Three Hydrangea Cultivars

To further identify the key genes responsible for the color differences observed in the sepals of three distinct hydrangea varieties, we conducted RNA-sequencing (RNA-seq) and de novo assembly on the sepals of these varieties. RNA-seq data were utilized to investigate the expression profiles of the essential genes involved in anthocyanin formation among the sepals of three hydrangea cultivars. PCA, Venn diagram, and clustering heat map analysis, performed with three biological replicates, revealed minimal variation within each sample group, signifying robust data accuracy, while also demonstrating a marked distinction between groups (Figure 3). PCA indicated minimal variation within each sample, signifying strong data repeatability, whereas a substantial difference was observed between groups (Figure 3A).

PCA and heatmap clustering showed little difference within each group of samples, indicating good data reproducibility, while there was a significant difference between groups (Figure 3A,C). RNA-seq data produced 6.63–7.36 GB of clean bases, with an error rate below 0.03%, a Q_20_ value exceeding 96.9%, a Q_30_ value surpassing 91.9%, and a GC content between 44.43% and 44.86% (Appendix A). Appendix A details the length and quantity of assembled transcripts and genes. A total of 201,506 transcripts and 105,743 unigenes were identified, with mean lengths of 1030 and 1317, respectively. The results of gene annotation reveal that the KEGG, NR, Swiss-Prot, Trembl, KOG, GO, and Pfam databases annotated 45,473, 59,231, 43,012, 59,267, 35,055, 51,107, and 39,956 genes, respectively, culminating in a total of 105,743 annotated genes (Appendix A). To acquire detailed gene function information, 56.01% (59,231) of the unigenes were annotated with their putative functions in the Nr database. The four species with the highest BLASTx hits were *Nyssa sinensis* (19,279 hits, 32.55%), *Camellia sinensis* (5759 hits, 9.72%), *Camellia lanceoleosa* (3104 hits, 5.24%), and *Actinidia chinensis* var. *chinensis* (2972 hits, 5.02%) (Appendix A).

Differential expression analysis was conducted using DESeq, comparing samples from three biological replicates of ‘DB’, ‘LB’, and ‘GB’ hydrangeas. A total of 36,784 unigenes exhibiting significantly altered expression levels (DESeq2 padj < 0.05 | log2 (fold change) > 1) were identified. A Venn diagram illustrates the distribution of these DEGs across the three comparison groups: ‘DB’ versus ‘LB’, ‘GB’ versus ‘DB’, and ‘GB’ versus ‘LB’. The Venn diagram revealed that 1223 DEGs were shared among the comparisons of ‘GB’ versus ‘LB’, ‘GB’ versus ‘DB’, and ‘LB’ versus ‘DB’ (Figure 3B). A total of 13,112, 13,898, and 9774 DEGs were identified for the comparisons of ‘DB’ versus ‘LB’, ‘GB’ versus ‘DB’, and ‘GB’ versus ‘LB’, respectively (Appendix A). Subsequently, we analyzed the DEGs among the three samples exhibiting distinct colors in a heat map (Figure 3C). To analyze the expression patterns of DEGs, we used R to standardize the FPKM expression levels. Subsequently, k-means clustering was employed to group genes exhibiting analogous expression patterns across three varieties, suggesting potential functional similarities (Figure 3D). This indicates that the expression of numerous genes varied significantly across the different cultivars.

### 2.4. GO Annotation and KEGG Pathway Analysis of Hydrangea Cultivars

Gene Ontology classification analysis indicated that numerous genes associated with ‘biological process’, ‘cellular component’, and ‘molecular function’ were present among the differentially expressed genes (Figure 4). In the comparison of the blue-type color cultivars ‘DB’ and ‘LB’, the most significantly enriched biological process was identified as the ‘phosphorelay signal transduction system’ (GO:0000160), the cellular process as the ‘intrinsic component of plasma membrane’ (GO:0031226), and the molecular function as ‘enzyme inhibitor activity’ (GO:0004857) (Figure 4A,C). In a comparative analysis of the green-type colored cultivar ‘GB’ with ‘DB’ and ‘LB’, the GO classification revealed that the DEGs were grouped within the same GO terms for ‘biological process’, ‘cellular component’, and ‘molecular function’ (Figure 4E). These terms included ‘photosynthesis, light reaction’ (GO:0019684), ‘photosystem’ (GO:0009521), and ‘glucosyltransferase activity’ (GO:0046527), respectively. The results indicated that the difference between ‘GB’ and both ‘LB’ and ‘DB’ is similar.

We performed a KEGG pathway analysis to explore the biological function of DEGs, which includes a thorough assessment of intracellular metabolic pathways and the activities of unigenes. The 20 most significantly enriched route categories were identified and presented in Figure 4B,D,F after comparing ‘LB’ and ‘GB’ with ‘DB’. The pathways with the most representative unigenes in these three comparisons were ‘Glutathione metabolism’, ‘Biosynthesis of secondary metabolites’, ’Phenylpropanoid biosynthesis’, ‘Flavonoid biosynthesis’, ‘Plant hormone signal transduction’, and ‘Anthocyanin biosynthesis’. Notably, ‘Metabolic processes’ and ‘Biosynthesis of secondary metabolites’ were the two most enriched pathways in the comparisons of ‘GB’ with ‘LB’ and ‘GB’ versus ‘DB’, with 1594 and 929, and 1901 and 1076 DEGs, respectively. The paths convey that the color creation of ‘GB’ significantly differs from that of ‘LB’ and ‘LB’, offering vital insights for examining the specific processes and pathways involved in the color distinction of hydrangea sepals.

### 2.5. Transcription Factors Involved in Color Formation in Hydrangea

Transcription factors serve as key regulators in the growth and development of plants. Transcription factors, including MYBs and bHLHs in *Arabidopsis*, significantly influence flavonoid production [22,23]. We identified 460, 490, and 444 differentially expressed transcription factors (DETFs) in the comparisons of ‘LB’ vs. ‘DB’, ‘DB’ vs. ‘GB’, and ‘LB’ vs. ‘GB’ (Table 1). Additionally, 204 TFs were upregulated in ‘LB’ compared to ‘DB’, while 256 were downregulated. Likewise, 298 TFs were upregulated in ‘DB’ compared to ‘GB’, while 192 were downregulated. In the comparison of ‘LB’ and ‘GB’, 243 TFs were upregulated while 201 TFs were downregulated. Overall, the following transcription factors were identified as having a significant impact: WRKY (4.78%), MYB (4.68%), C3H (4.38%), AP2/ERF (4.16%), bHLH (3.73%), C2H2 (3.63%), NAC (3.4%), FAR1 (3.4%), and SET (3.4%). In addition, the remaining TFs account for approximately 63.48% (Appendix A). Among them, *ERF*, *NAC*, *WRKY*, *MYB*, *bHLH*, and *bZIP* transcription factors were paramount, indicating their significant function in modulating color biosynthesis genes and their essential contribution to hydrangea color creation.

### 2.6. Differential Expression of Structural Genes Associated with Anthocyanin Biosynthesis Pathway

Gene expression in the anthocyanin biosynthesis pathway was compared across the three cultivars, identifying 51 DEGs associated with anthocyanin biosynthesis, as illustrated in Figure 5. Anthocyanins are a subset of flavonoids that share an identical synthetic pathway with other flavonoids and are produced by branched production processes originating from dihydrokaempferol. Figure 5 illustrates that, upstream of the synthesis pathway elements, the majority of the *CHI*, *CHS*, and *DFR* genes had elevated expression levels in ‘DB’ and ‘LB’, while demonstrating reduced expression in ‘GB’, thereby supplying substrates for the production of downstream anthocyanin compounds. Following the synthesis pathway, the expression levels of *CHS* (cluster-26710.2), *CHI* (cluster-47889.0), *F3H* (cluster-28792.0), *F3′5′H* (cluster-39474.0), *DFR* (cluster-46471.3), *ANS* (cluster-51681.0), *BZ1* (cluster-57331.0), and *3AT* (cluster-38007.2) progressively diminished from ‘DB’ to ‘GB’, correlating with the delphinidin content. Furthermore, *CHS* (cluster-26710.2) and *DFR* (cluster-46471.8) exhibit expression alone in ‘DB’, which was not detected in ‘LB’ and ‘GB’. In the anthocyanin biosynthesis process, the enzyme DFR (Dihydroflavonol 4-reductase) plays a crucial regulatory role by converting different dihydroflavonols (including dihydrokaempferol, dihydroquercetin, and dihydromyricetin) into colorless anthocyanidins (like leucodelphinidin). This is a vital stage in the production of anthocyanins, which are responsible for the vibrant hue of plants and contribute to how plants adapt to adverse conditions in their surroundings. The results demonstrated that variations in the expression of structural genes in hydrangea sepals may result in differing biosynthesis and accumulation of anthocyanins, especially delphinidins, leading to diverse sepal colors.

### 2.7. Differential Expression of Structural Genes Linked to Chlorophyll Biosynthesis Pathway

The macromolecules (tetrapyrroles and magnesium) that give plants their green hue are called chlorophylls (Chls). Chlorophylls, a type of vital photosynthetic pigment, are present in nearly all plants and primarily contribute to the green coloration of flowers [24,25]. The number of genes (31 DEGs) linked to chlorophyll synthesis has been significantly elevated in the hydrangea cultivar ‘GB’, characterized by its green flower sepals (Figure 6). DEGs associated with chlorophyll biosynthesis were *HemA*, *HemB*, *HemL*, *HemC*, *HemD*, *HemE*, *HemF*, *HemY*, *chlD*, *chlH*, *chlM*, *chll*, *CRD1*, *DVR*, *PORA*, *CAO*, *NOL*, and *HCAR*. *HemA* (cluster-497229.2 and cluster-51775.0), *HemE* (cluster-27864.0 and cluster-33216.0), *HemF* (cluster-24014.0), *HemY* (cluster-54852.0), *chlD* (cluster-50382.0), *chll* (cluster-35667.0), *chlM* (cluster-18472.0), *CRD1* (cluster-46309.0), *PORA* (cluster-31306.0 and cluster-31306.1), and *HCAR* (cluster-30576.0) exhibited a progressive increase in expression from cultivar ‘DB’ to ‘LB’, peaking in ‘GB’. Curiously, *NOL* (cluster-42095.3) showed higher expression in ‘DB’ and ‘LB’ compared to ‘GB’. Likewise, *NOL* (cluster-42095.0 and cluster-42095.1) showed maximal expression in ‘LB’ compared to ‘DB’ or ‘GB’. NOL converts chlorophyll b into 7-hydroxymethyl chlorophyll a, which is subsequently transformed into chlorophyll a by *HCAR* enzyme genes. Notably, *PORA* (cluster-31306.0 and cluster-31306.1) and *CRD1* (cluster-46309.0) showed maximal expression in the green blue hydrangea cultivar (‘GB’) compared to other chlorophyll biosynthesis genes, implying their putative key role in green color formation in the sepals. Furthermore, a significant number of chlorophyll biosynthesis genes had the highest expression in ’GB’ relative to the other cultivars.

### 2.8. Validation of DEG Profiling via qRT-PCR

To validate the RNA-seq, ten differentially expressed genes linked to anthocyanin synthesis were chosen randomly for qRT-PCR analysis (Appendix A). The genes include *DFR*, *CHI*, *UGFT*, *P450*, and *CYP73A*, as well as the transcription factors *MYB*, *NAC*, *WRKY*, and *AP2/ERF* (Figure 7). The expression profile of genes measured by qPCR closely aligned with the RNA-seq data values. The expression levels of *NAC/P450/MYB/CYP73A* were highest in the cultivar ‘LB’ compared to the other cultivars. The expression levels of *WRKY/AP2/ERF* were highest in ‘GB’, while *UTG75C/CHI/DFR* exhibited maximal mRNA transcript levels in ‘DB’. The qRT-PCR experiments confirmed the reliability of the RNA-seq-derived gene expression profiles.

## 3. Discussion

The coloration of flowers is predominantly influenced by the chemical composition of flavonoids, particularly the subclass known as anthocyanins. Flavonoids, anthocyanins, betalains, carotenoids, and chlorophylls are recognized for their significant role in influencing plant color pigmentation [18,26,27,28]. The color of flowers is a significant characteristic of horticultural crops, exemplified by *H. macrophylla* [1]. It impacts economic value, visual appeal, aesthetics, and quality. There are over 400 cultivars of *H. macrophylla* exhibiting a variety of flower color phenotypes [29,30], yet the modulatory mechanism behind color formation among cultivars is less known. A novel mutation with green flowers has been found, offering an unprecedented chance to examine sepal coloration in *H*. *macrophylla*.

This study employed a colorimeter to analyze the coloration of three distinct *H. macrophylla* sepals labeled ‘LB’, ‘GB’, and ‘DB’. The ‘DB’ exhibits the highest value of *a** alongside the lowest value of *b**, while the ‘GB’ shows the highest value of *b** and the lowest value of *a** (Figure 1B–D). The * value correlated with increased anthocyanin levels in the ‘DB’ cultivar. In previous investigations, a colorimeter was utilized to measure and analyze gerbera and caladium petals, effectively classifying them into distinct color groups. The analysis revealed significant variations in the total carotenoids and total anthocyanins among different color series [31,32]. Analysis showed that ‘GB’ sepals had significantly higher chlorophyll content than ‘DB’ and ‘LB’, approximately 2–3 times greater (Figure 1E). Studies showed that chlorophyll content causes green pigmentation, playing a key role in the coloration of ‘GB’ sepals. These studies suggest that colorimetric detection effectively analyses flower color phenotypes.

Metabolomics facilitates an in-depth exploration of the intricate biochemistry underlying flower coloration, thereby enhancing researchers’ comprehension of flower color formation in ornamental plants [33,34]. Targeted metabolomics highlights the investigation of particular classes of metabolites with enhanced selectivity and precision [35]. Herein, UPLC-MS/MS was employed to profile the sepals of three *H*. *macrophylla* genotypes to discover flavonoid and anthocyanin substances that may explain the blue coloration in the ‘DB’ and ‘LB’ genotypes relative to the ‘GB’ genotype (Figure 2). These cultivars included 47 anthocyanin types, with blue sepals accumulating more than light blue or green ones. In the ‘LB’ and ‘DB’ sepals of *H. macrophylla*, the anthocyanin components are comparable, with the primary anthocyanin (delphinidin-3-*O*-glucoside) content being twice as high in the ‘DB’ compared to the ‘LB’. The elevated anthocyanin concentration in petals and sepals can intensify the coloration of flowers. For instance, anthocyanins are absent in the white flower mutants of *Gentiana triflora*, while they accumulate in significant quantities in *G. triflora* blue flowers, predominantly delphinidin [36,37]. A previous study of *H*. *macrophylla* ‘Endless Summer’ showed that carotenoids and flavonoids in blue sterile flowers decrease from bud to full bloom while anthocyanin increases to its peak [38,39]. Our investigation found that ‘DB’ had a similar effect, with anthocyanins at 1512 µg/g and reduced flavonoids. Likewise, in *Gloriosa superba* ‘Rothschildiana’ anthocyanin (cyanidin-3,5-*O*-diglucoside, cyanidin-3-*O*-glucoside, pelargonidin-3-*O*-glucoside, and pelargonidin-3,5-*O*-diglucoside) were the major content in the floral petals, resulting in intense petal color [40].

Like other species, anthocyanin was concentrated in the sepals of *H*. *macrophylla*. This study identified six pigment types that were exclusive to specific-colored samples, suggesting their significant significance in color formation. Twenty-five overlapping DAMs were identified across all cultivars, which may contribute to the variation in sepal hues (Figure 2B,C). Analysis showed significant variation in delphinidin-3-*O*-glucoside and delphinidin-3-*O*-sambubioside levels among ‘DB’, ‘LB,’ and ‘GB’ (Figure 2D). These results lend credence to the claims made by Yoshida et al. [7,15] and Zhang et al. [41] that the principal component of blue-violet flowers is delphinidin-3-*O*-glucoside. Delphinidin was mostly associated with the development of blue blooms. Several blue flowers, such as Cineraria, delphinium, Rhododendron, campanula, Petunia, *Salvia miltiorrhiza*, and *Glycine soja*, have been identified as containing delphinidin-3-*O*-glucoside, indicating that this compound may be responsible for the development of blue coloration in these flowers [42,43,44,45,46]. The study found delphinidin-3-*O*-glucoside in all colored flowers, especially in blue varieties, notably in ‘DB’ (Figure 2E,F). The ‘GB’ green flowers with a blue edge contained delphinidin-3-*O*-glucoside. This study identified four delphinidin derivatives: delphinin-3-*O*-glucoside, delphinidin-3-*O*-sambubioside, delphinidin-3,5-*O*-diglucoside, and delphinidin-3-*O*-rutinoside-7-*O*-glucoside, which are essential for the blue color formation in *H*. *macrophylla*. Malvidin derivatives are formed from delphinidin derivatives, modified into malvidin by the gene *OMT*.

The synthesis of anthocyanin, chlorophyll, and carotenoid metabolites is governed by multiple structural genes associated with these mechanisms. We identified 51 structural genes involved in anthocyanin biosynthesis, including *CHS*, *F3*′H, *F3′5′H*, *DFR*, *ANS*, *BZ1*, and *3AT* (Figure 5). It is noteworthy that *F3′5′H*, *DFR*, *ANS*, *BZ1*, and *3AT* genes exhibited a high level of expression in the cultivar ‘DB’, playing a key role in anthocyanins biosynthesis during the later stages [47]. In anthocyanin biosynthesis, *DFR* (dihydroflavonol 4-reductase), *ANS* (anthocyanidin synthase), and *BZ1* (anthocyanidin 3-O-glucosyltransferase) are essential genes involved in the later stages of anthocyanin production. DFR catalyzes the conversion of dihydroflavonols to leucoanthocyanidins, while ANS facilitates the transformation of leucoanthocyanidins into anthocyanidins [18,44]. In this study, we identified 12 *DFR* genes among the DEGs. Notably, the expression of the *DFR* gene (cluster-46471.3) was significantly upregulated in the sepals of both ‘DB’ and ‘LB’ compared to ‘GB’, exhibiting 11.22-fold and 8.74-fold increases, respectively. A similar study was conducted on the Iris flower, where two *DFR* genes (*IlDFR* and *IlaDFR*) were cloned from *I. laevigata* (deep blue) and *I. lactea* (light blue), respectively. The data showed that *DFR* genes regulate delphinidin synthesis, essential for Iris flowers’ blue coloration [48]. A higher anthocyanin content was associated with higher expression levels of *FvDFR30*, *FvDFR54*, and *FvDFR56* genes in strawberries, indicating that *DFR* genes may play a pivotal role in the synthesis of anthocyanins in strawberries [49].

The *3AT* (3-*O*-acetyltransferase) gene plays a significant role in anthocyanin metabolism, particularly in the formation of flower color. The *3AT* gene catalyzes the acetylation of anthocyanidin-3-glucosides, leading to the production of 3-*O*-acetylated anthocyanins, which contribute to the deepening of flower color [50]. In this study, one *3AT* gene (cluster-38007.2) was related with 10 anthocyanins and expressed more in ‘DB’ than ‘LB’ or ‘GB’. Among them, pelargonidin-3,5-*O*-diglucoside exhibited a negative correlation in ‘DB’, ‘LB’, and ‘GB’. The remaining seven DAMs accumulated to a greater extent in ‘DB’ compared to ‘LB’ or ‘GB’. Additionally, cyanidin-3-*O*-5-*O*-(6-*O*-coumaroyl)-diglucoside and cyanidin-3,5-O-diglucoside were absent in ‘LB’. A prior study demonstrated the essential function of *3AT*. The 3AT enzyme in *Perilla frutescens* can transfer p-coumaric and caffeic acids from their CoA esters to the 3-glucosyl moiety of delphinidin, cyanidin, and pelargonidin 3-glucosides. The research indicated that the 3AT enzyme results in a deep blue appearance of the flower hue due to acetylation [50]. In the anthocyanin-rich ‘Zijuan’ tea plant, 3AT was elevated along with other biosynthetic genes and TFs and was found to increase and alter anthocyanins [50]. These findings suggest that these genes may associated with sepal color variability in *H. macrophylla*.

Chlorophyll is a fundamental element in the floral coloration. This study identified structural genes associated with chlorophyll production. Thirteen of these genes expressed higher in the green mutant ‘GB’ than in the blue types ‘DB’ and ‘LB’. Numerous crucial genes were implicated in the extensively researched chlorophyll production pathway [51,52]. HemA catalyzes the conversion of glutamyl-tRNA to L-Glutamate 1-semialdehyde, serving as the primary enzyme in chlorophyll production and regulation during de-etiolation [53]. The protochlorophyllide reductase (POR), the enzyme facilitating the photoreduction of protochlorophyllide to chlorophyllide, is essential throughout the greening phase. In the ‘GB’ cultivar, the expression levels of *HemA*, *HemE*, *HemF*, *CRD1*, and *PORA* were elevated compared to ‘LB’ and ‘DB’, indicating a heightened rate of chlorophyll production and accumulation. Prior research on green chrysanthemum and carnation flowers found that pale green carnation petals had more chlorophyll than non-green carnation petals [54]. Similar observations were made regarding the transformation of lily petals from green to white, highlighting the link between the greening of hydrangea sepals and chlorophyll synthesis.

In addition to structural genes, TFs significantly contribute to anthocyanin biosynthesis, including MYB, bHLH, WD proteins, and WRKY proteins [55,56,57]. In *Medicago truncatula*, bHLH and WD40 proteins regulate the production of anthocyanins and proanthocyanidins [58]. Similarly, in *Syringa oblata*, SoNAC72-SoMYB44/SobHLH130 modulates floral pigmentation by directly interacting with the promoters of structural anthocyanin biosynthesis genes [59]. Prior research, including ours, displays that AN modulates the expression of *CHS*, *DFR*, and *AAT1*. Herein, we found several TFs linked to floral color, including MYB, ERF, bHLH, WRKY, and NAC. These transcription factors were expressed significantly and in higher quantities than others, especially in the ‘DB’ cultivar (Table 1). Our findings correspond with previous research indicating that transcription factors, including ERF, MYB, and bHLH, are pivotal in secondary metabolism, including floral pigmentation [57,60,61]. Particularly, in the comparison of ‘DB’ and ‘GB’, the quantity of upregulated TFs such as ERF, NAC, MYB, WRKY, and bHLH exceeded that of downregulated factors, indicating their essential function in the production of floral color.

## 4. Materials and Methods

### 4.1. Plant Materials

The experimental materials used in this study comprised three varieties of *H. macrophylla*. The first variety, designated as ‘Bo Da Lan’, exhibits a deep blue coloration (abbreviated as ‘DB’). The second cultivar, named ‘Qian Lan’, displays a light blue hue (referred to as ‘LB’). Additionally, a bud mutation strain is characterized by predominantly green sepals with a blue edge (abbreviated as ‘GB’). The ‘DB’ variety has emerged as a primary cultivation type of hydrangea, characterized by early flowering, larger inflorescence size, compact flowers, deeper coloration, and broad adaptability. The ‘LB’ variety exhibits smaller inflorescences and a lighter blue hue than ‘DB’, but demonstrates greater plant height. ‘GB’ is a bud mutation strain derived from ‘LB’. The primary distinction lies in the sepal color, with ‘GB’ exhibiting a predominantly green hue, uncommon within the hydrangea group. For every cultivar, three or four biological replicas were collected.

This study used plants cultivated in the greenhouse at the nursery of Kunming Yang Chinese Rose Gardening Co., Ltd. (Kunming, China). The corymbs were collected from early May to mid-August during the blossoming period. The plants experienced analogous natural conditions. To investigate the color formation of these three cultivars, combined transcriptomic and metabolomic techniques were employed to elucidate the changes in metabolites and the relationship between metabolites and gene expression associated with floral sepal color variation. All experiments were performed with three biological replicates.

### 4.2. Floral Sepal Color Measurement

During the peak blooming period, the sepal hues were documented using the Royal Horticultural Society Color Chart with a white background and consistent lighting settings. Additionally, a CS-210 precision colorimeter (Hangzhou Caipu Technology, Hangzhou, China) was employed to measure the color variables of the identical flower sepals. The characteristics of the CIE*L*a*b** hue coordinate system (chromatic components, *a** and *b**; lightness, *L**) were determined. The CIELAB nonlinear modification of the RGB color space makes the real distance between two colors equal to their Euclidean distance (for distances less than 10 units). When working with color image processing techniques, CIELAB often yields excellent outcomes [62]. Each cultivar was measured five times as a replica.

### 4.3. Total Chlorophyll Content Measurement

Fresh sepals from three cultivars were collected and thoroughly washed with distilled water to remove surface contaminants. A small quantity of the sepals (about 0.2 g) was placed in a mortar, pounded with a pestle, and then 10 mL of 95% ethanol was introduced to enhance the extraction process. The resulting mixture was centrifuged at 5000× *g* for 10 min to separate the soluble pigments from the insoluble plant debris. The supernatant containing the extracted chlorophyll was collected and used for further analysis. The chlorophyll content was determined using a spectrophotometric method. The absorbance of the extract solution was measured at wavelengths of 645 nm and 663 nm using a UV-Vis spectrophotometer. Total chlorophyll content was determined by combining the amount of chlorophyll a and chlorophyll b. To acquire a single average read, three separate corymb reads were utilized to acquire a single average read. The data was collected from five individual plants.

### 4.4. RNA Extraction and RNA-Seq

RNA extraction was performed using the TransZol Up kit (Yun Geng Biology, Kunming, China), followed by reverse transcription of RNA to cDNA using the HiScript III RT SuperMix for qPCR (including gDNA wiper). The qPCR was conducted with the ChamQ Universal SYBR qPCR Master Mix. Sequencing adapters were attached to double-stranded cDNAs, followed by amplification and purification. Metware Biotechnology Co., Ltd. (Wuhan, China) sequenced the cDNA libraries using Illumina sequencing. The assay was performed in three biological replicates.

### 4.5. De Novo Assembly and Functional Annotation

Raw sequencing data were preprocessed using Trimmomatic v0.39 [63] to remove adapter sequences, low-quality bases (Q < 20), and reads shorter than 50 bp. Pure reads GC content was computed, and FastQC produced the Q_20_ and Q_30_ values to evaluate base quality. High-quality clean reads were obtained through this filtering process. These clean reads were then assembled into Unigenes using Trinity v2.13.2 [64]. The assembled Unigenes were annotated by aligning them against multiple databases, including Pfam, KEGG, Nr, Swiss-Prot, GO, KOG/COG, and Trembl. Transcription factor (TF) analysis was conducted using iTAK [65]. Gene expression levels were quantified using the FPKM score, and differentially expressed genes (DEGs) between samples were identified using DESeq2 [66]. DEGs were defined as those with a |log2Fold Change| ≥ 1 and a false discovery rate (FDR) < 0.05. Functional and network enrichment analyses of DEGs were performed using the Eggnog-mapper [67] tool, with a *p*-value < 0.05 considered indicative of significant enrichment.

### 4.6. UPLC-MS/MS for Targeted Metabolite Analysis

#### 4.6.1. Chemicals and Reagents

HPLC-grade methanol (MeOH) was acquired from Merck (Darmstadt, Germany), formic acid from Sigma-Aldrich (St. Louis, MO, USA), and hydrochloric acid from Xinyang Chemical Reagent (Changsha, China). MilliQ water (Millipore, Bradford, PA, USA) was utilized in all investigations, and standards were acquired from isoReag (Shanghai, China). Standard stock solutions were produced at 1 mg/mL concentration in 50% MeOH. All stock solutions were kept at −20 °C. Before analysis, the stock solutions were diluted with 50% MeOH to generate working solutions.

#### 4.6.2. Sample Preparation and Extraction

The sample was freeze-dried, pulverized (30 Hz, 1.5 min), and preserved at −80 °C until required. Fifty milligrams of powder were weighed and extracted using 0.5 mL of a methanol/water/hydrochloric acid solution (500:500:1, *V*/*V*/*V*). The extract was vortexed for 5 min, subjected to ultrasonic for 5 min, and centrifuged at 12,000× *g* at 4 °C for 3 min. The residue was further extracted by repeating the abovementioned methods under identical conditions. The resulting supernatants were obtained and filtered using a 0.22 μm membrane filter (Anpel) before LC-MS/MS analysis.

#### 4.6.3. UPLC Conditions

A UPLC-ESI-MS/MS system (UPLC: ExionLC™ AD, MS: Applied Biosystems 6500 Triple Quadrupole, https://sciex.com.cn/) was employed to analyze the samples. The analytical conditions were as follows: UPLC: column, water; ACQUITY BEH C18 (1.7 µm, 2.1 mm × 100 mm); solvent system, water (0.1% formic acid), methanol (0.1% formic acid); gradient program, 95:5 *V*/*V* at 0 min, 50:50 *V*/*V* at 6 min, 5:95 *V*/*V* at 12 min, hold for 2 min, and 95:5 *V*/*V* at 14 min; hold for 2 min; flow rate: 0.35 mL/min; temperature: 40 °C; injection volume: 2 μL.

#### 4.6.4. ESI-MS/MS Conditions and Metabolites Quantification

Linear ion trap (LIT) and triple quadrupole scans were obtained using a triple quadrupole-linear ion trap mass spectrometer (QTRAP), specifically the QTRAP^®^ 6500+ LC-MS/MS System. This system features an ESI Turbo Ion-Spray interface, operates in positive ion mode, and is managed by Analyst 1.6.3 software (Sciex). The parameters for the ESI source operation were outlined as follows: Ion source: ESI+; source temperature: 550 °C; ion spray voltage (IS): 5500 V; curtain gas (CUR) was set at 35 psi. Anthocyanins were examined utilizing timed multiple reaction monitoring (MRM). The collection of data was conducted using Analyst 1.6.3 software (Sciex). Multiquant 3.0.3 software (Sciex) was employed for the quantification of all metabolites. The parameters for the mass spectrometer, including declustering potentials (DP) and collision energies (CE) for specific MRM transitions, were optimized further for DP and CE. A designated group of MRM transitions has been recorded for all times based on the metabolites eluted during that time frame. To maximize the precision of both qualitative and quantitative analyses, chromatographic peaks of the analytes were adjusted in various samples according to the retention time and peak shape data of anthocyanin standards in Metware database (MWDB). The subsequent approach was employed to determine and quantify anthocyanins. The concentration value (ng/mL) of every sample was determined by replacing the integrated peak area with the standard curve equation. The absolute Log2FC (fold change) was used to determine whether metabolites were significantly regulated across the groups. The subsequent formula was employed to quantify the metabolite levels in the sample (µg/g): The concentration of metabolites in the sample (µg/g) is calculated using the formula: Levels = c × V/1,000,000/m, where c represents the concentration, V denotes the extraction solvent volume (µL), and m indicates the mass of the sample (g).

#### 4.6.5. Identification of Differential Accumulation Metabolites

Differential accumulated metabolites (DAMs) were screened based on the criteria of a VIP value greater than 1, a Log2FC of at least 1, and a *p*-value less than 0.05.

### 4.7. Quantitative Real-Time PCR

The qRT-PCR assay was conducted on chosen DEGs linked with the anthocyanin and chlorophyll production pathways and key transcription factors. Total RNA extraction and cDNA synthesis were performed as previously described [68,69]. Primer pairs are listed in the Appendix A. Actin was used as an endogenous control. Gene expression levels were determined using the delta CT technique [70]. The experiment was performed in triplicate.

### 4.8. Statistical Analysis

To compare the mean values, a one-way analysis of variance (ANOVA) was performed on the data using SPSS 16.0 (SPSS, Inc., Chicago, IL, USA) and by employing Duncan’s multiple range tests at a significance level of *p* < 0.05. The outcomes of hierarchical cluster analysis (HCA) for samples and metabolites were displayed as heatmaps accompanied by dendrograms. HCA and K-means were performed using the built-in functions in R version 4.4.2.

## 5. Conclusions

This study examined the anthocyanin metabolites and their corresponding genes in three hydrangea cultivars: ‘DB’, ‘LB’, and ‘GB’. A total of 47 anthocyanin metabolites were identified, predominantly consisting of delphinidin, cyanidin, and pelargonidin derivatives, with an observed increase in total content in deep blue flowering shade (‘DB’). Deciphering the transcriptome has revealed critical DEGs linked to pathways involved in the translation of hormone signals and the production of anthocyanins and chlorophylls, the two primary pigments in flower sepals. This elucidated the molecular mechanisms behind anthocyanin accumulation and production in hydrangea flower sepal pigmentation, enhancing our understanding of the biochemical basis of flower sepal color in *H*. *macrophylla*.

## Figures and Tables

**Figure 1 plants-14-00742-f001:**
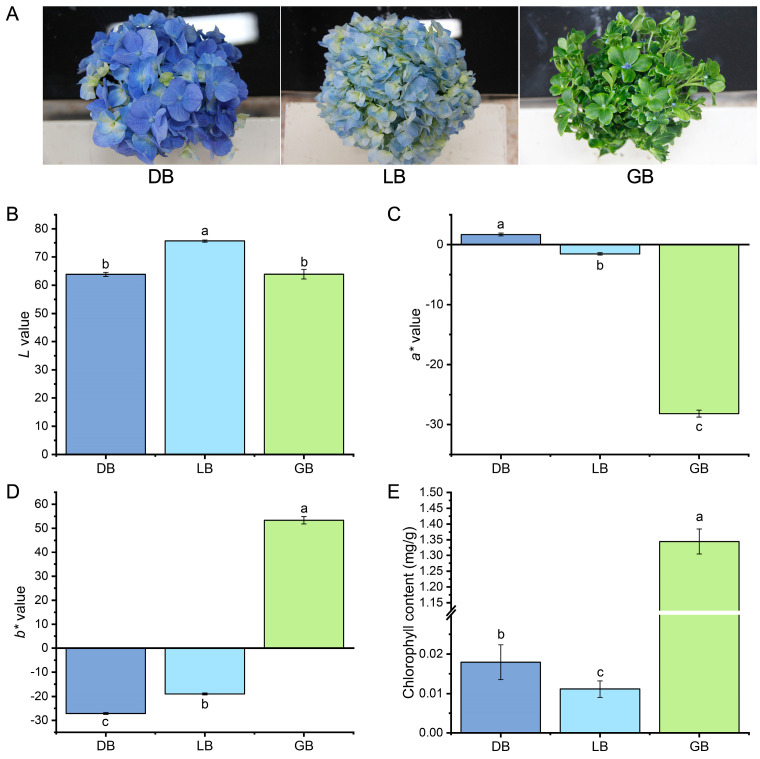
Phenotypic illustration, color, and chlorophyll content of three hydrangea cultivars. (**A**) Pictorial view of three hydrangea cultivars. (**B**) *L**, (**C**) *a**, (**D**) *b**, and (**E**) total chlorophyll content in hydrangea cultivars. Total chlorophyll content was obtained by summing the chlorophyll a and b concentrations. Data is denoted as the mean ± SEM (n = 3). Disparities were determined using least significant difference (LSD) at *p* < 0.05, with substantially different groups denoted by distinct letters (a, b, c). All measurements were performed in triplicate to ensure reproducibility and accuracy.

**Figure 2 plants-14-00742-f002:**
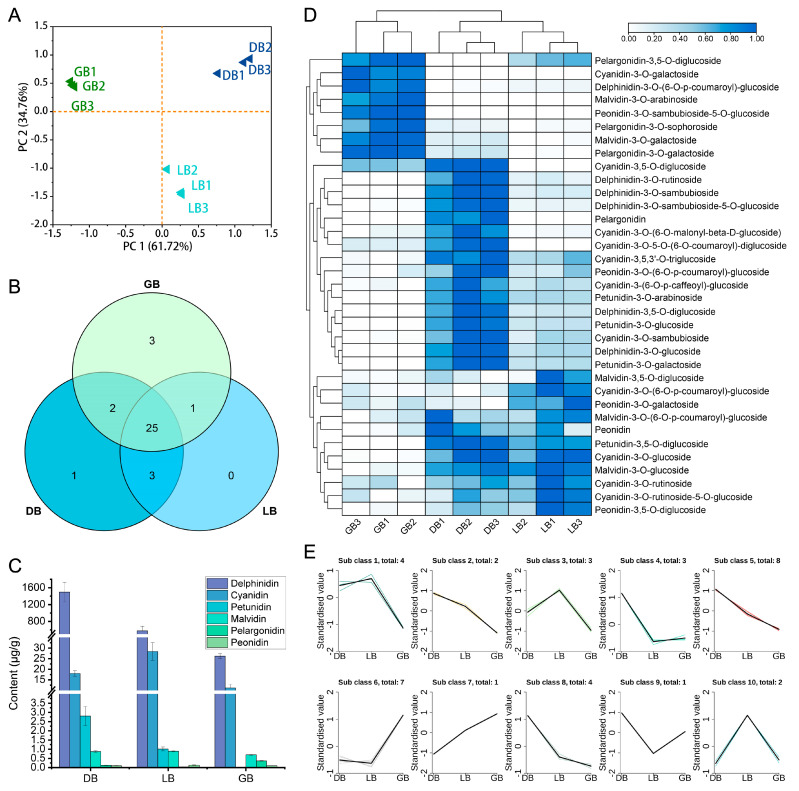
Analysis of anthocyanin metabolites in three hydrangea cultivars. (**A**) PCA score plot, (**B**) Venn diagram, (**C**) various anthocyanin pigments, (**D**) Heat map, and (**E**) k-means cluster analysis of anthocyanin-related metabolites in hydrangea cultivars. GB1-3, DB1-3, and LB1-3 denote the biological replicates. Diverse colored lines denote cumulative patterns of distinct metabolites in each cultivar. A heatmap was generated via normalized data. Colors indicate scaled expression values, with blue for the highest expression and white for low expression levels.

**Figure 3 plants-14-00742-f003:**
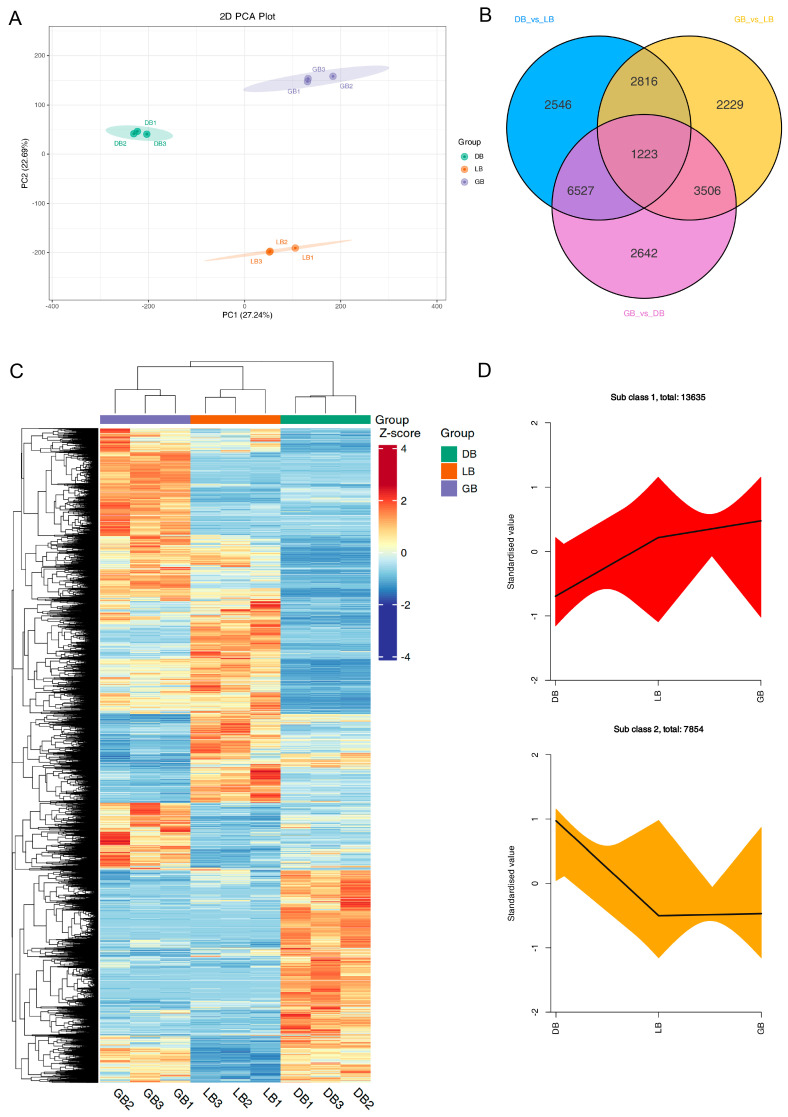
RNA sequence and DEGs analysis of three hydrangea cultivars. (**A**) PCA score plot, (**B**) Venn diagram, (**C**) Heatmap of differentially expressed genes (DEGs), and (**D**) k-means cluster analysis of DEGs in hydrangea cultivars. GB1-3, DB1-3, and LB1-3 denote the biological replicates. Diverse colored lines denote cumulative patterns of distinct metabolites in each cultivar. The heat map displays DEGs relative expression values (row z-score of normalized FPKM) in three hydrangea cultivars. Colors indicate scaled expression values, with red for the highest expression and blue for low expression levels.

**Figure 4 plants-14-00742-f004:**
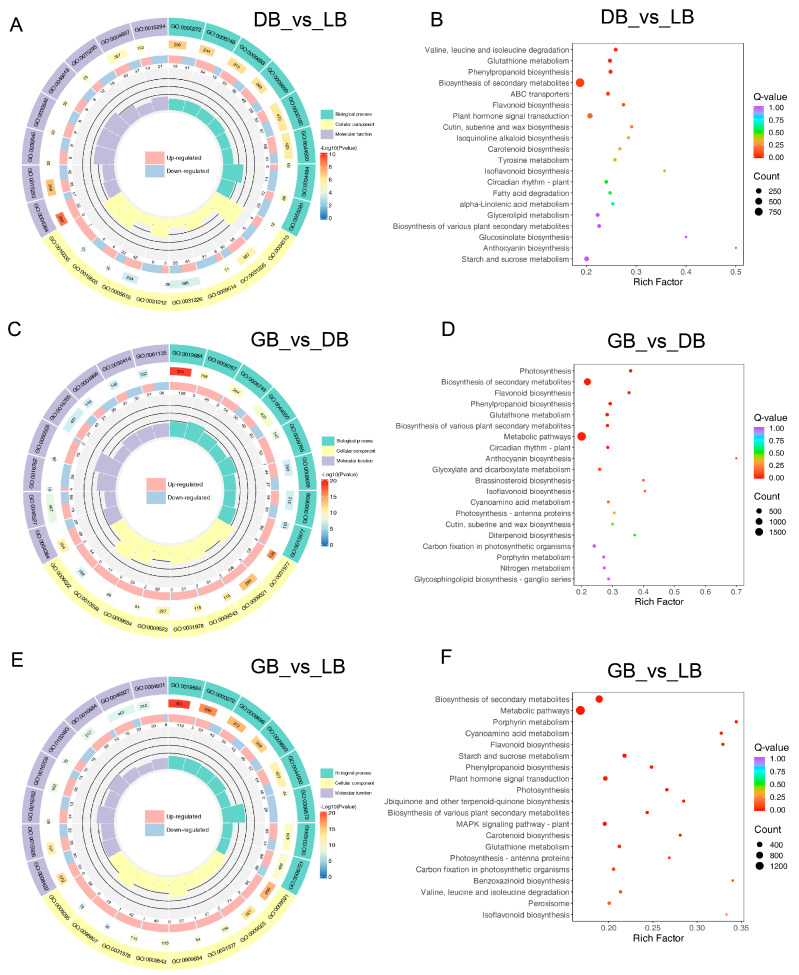
GO enriched and KEGG pathway of DEGs in three hydrangea cultivars. The size of the dots indicates the number of DEGs in the corresponding pathways. Gene ontology classification among (**A**) DB vs. LB, (**C**) GB vs. DB, and (**E**) GB vs. LB comparisons. KEGG analysis of DEGs among the (**B**) DB vs. LB, (**D**) GB vs. DB, and (**F**) GB vs. LB comparisons. KEGG category enrichment was determined by employing the R programming language along with the Cluster, Biobase, and Q-value packages (*p* ≤ 0.01).

**Figure 5 plants-14-00742-f005:**
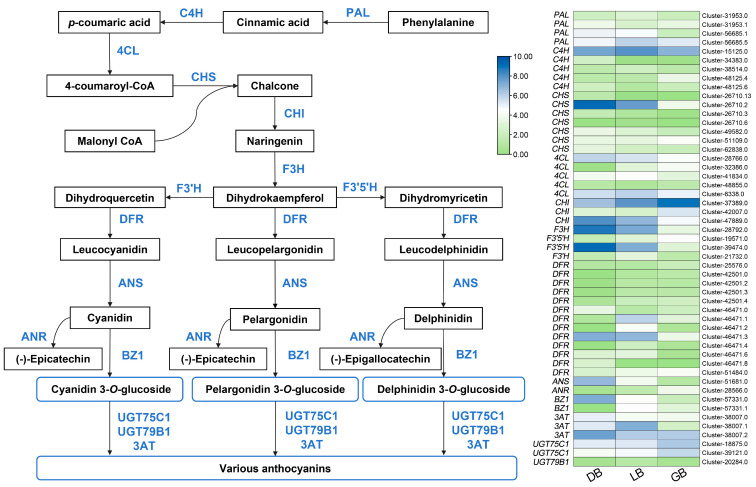
Key structural genes related to the anthocyanin biosynthesis pathway. The expression profiles of these genes in three hydrangea cultivars are illustrated on the right side of the figure. The green color signifies minimal expression, whereas blue represents maximal gene expression.

**Figure 6 plants-14-00742-f006:**
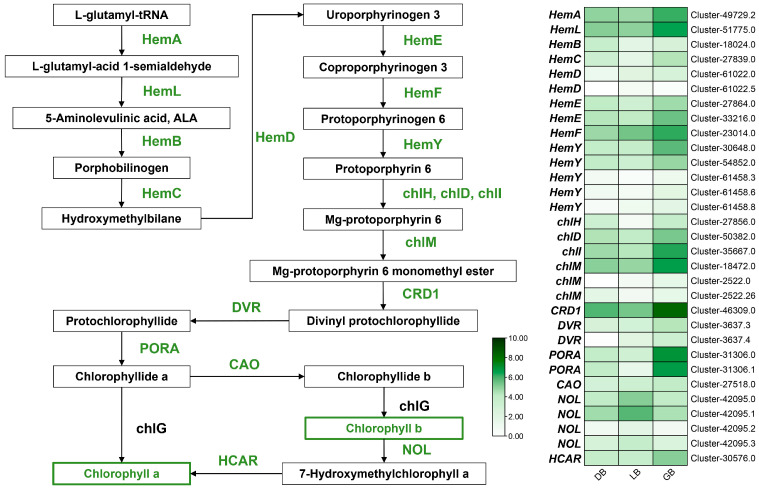
Key structural genes related to the chlorophyll biosynthesis pathway. The expression profiles of these genes in three hydrangea cultivars are illustrated on the right side of the figure. The green signifies the highest expression, whereas the white denotes the lowest gene expression.

**Figure 7 plants-14-00742-f007:**
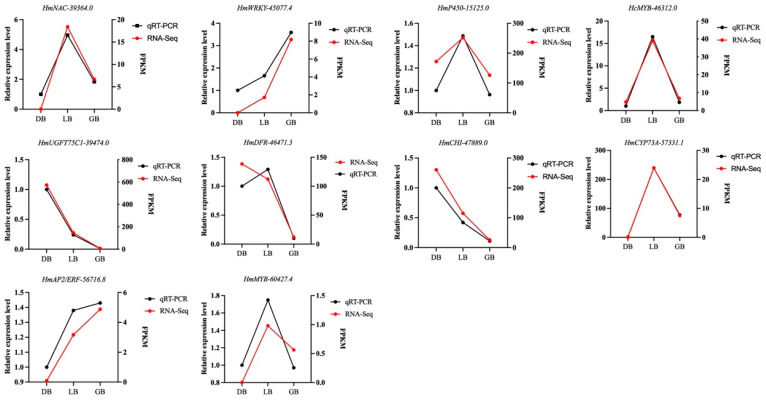
Validation of the expression profiles of ten randomly selected genes using qRT-PCR in together with RNA-seq data. Actin was used as an endogenous control. The black line signifies the qRT-PCR values, whereas the red line displays the RNA-seq results. Data is denoted as the mean ± SEM (n = 3).

**Table 1 plants-14-00742-t001:** List of TFs found in the DEGs of the hydrangea cultivars transcriptome dataset.

TFs	LB vs. DB		DB vs. GB		LB vs. GB	
	Up Regulated	Down Regulated	Up Regulated	Down Regulated	Up Regulated	Down Regulated
ERF	25	24	23	12	26	13
NAC	13	23	16	15	13	18
WRKY	16	13	24	12	23	9
MYB-related	9	20	17	15	7	22
bHLH	13	11	24	9	24	6
MYB	11	13	12	11	15	17
bZIP	8	15	7	11	5	22
C2H2	9	9	13	10	12	9
GRAS	11	14	13	10	8	5
FAR1	2	16	13	6	1	6
C2C2-Dof	7	6	7	7	8	4
C3H	5	12	11	5	3	2
LOB	7	5	4	7	1	4
G2-like	1	6	7	3	5	5
MADS-MIKC	3	1	8	4	8	3
TCP	4	4	6	3	5	4
GATA	6	2	6	3	7	1
HB-HD-ZIP	4	5	2	6	1	7
HB-other	2	9	8	2	3	1
B3-ARF	5	2	5	2	6	4
Others	43	46	72	39	62	39
Total	204	256	298	192	243	201

## Data Availability

The transcriptome raw data have been submitted to the SRA database of the NCBI (PRJNA1208947).

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
