# Peer review of "Function of Anthocyanin and Chlorophyll Metabolic Pathways in the Floral Sepals Color Formation in Different Hydrangea Cultivars"

_plants, 2025, doi:10.3390/plants14050742_

Round 1
Reviewer 1 Report
Comments and Suggestions for Authors
Dear authors
Below, you will find the observations and comments made during the review and evaluation of your manuscript, which should be considered for the updating of your manuscript.
Regards, reviewer
Overall evaluation
The submitted manuscript is extensive and addresses a relevant topic. However, it does not fully comply with the guidelines provided by Plants magazine. Here are the observations, strengths, weaknesses, and recommendations to improve the document.
1. Summary
The problem is not clearly defined. Absent methodology. Present results. Unidentified conclusion.
Recommendations: Write the abstract in a concise manner including problems, methods, key results, and conclusions. Reduce the length to fit the required format.
2. Introduction
The context is not clearly identified. The problem is absent. The theoretical or practical justification is not mentioned. There is the experimental design.
Recommendations: Including relevant information. Include a clear context, highlight the problem, and justify the research theoretically and practically.
3. Materials and Methods
The experimental design is not described. Variables and replicability are not identified. It is not mentioned whether the research requires the endorsement of the Ethics Committee.
Recommendations: Prepare this section including details of the experimental design, variables, controls, replications, and ethical approvals, when applicable. It is important to include information on legal and ethical permits related to the use of plant species, such as those required by national or international regulations (e.g., CITES, Nagoya Protocol). In addition, if the collection was carried out in protected areas or with associated traditional knowledge, permit numbers and compliance with local regulations must be indicated.
4. Results
Present statistical results. The use of tables and figures to summarise the findings is observed.
Recommendations: Check if the results are presented in order of importance and if the figures/tables complement the text efficiently.
5. Discussion
A synthesis of the results is found. Adequate comparison with other studies. An explanation of the mechanisms is included. Limitations are mentioned. Future hypotheses are proposed.
Recommendations: Reduce the length by eliminating redundancies and better organise the information. Focus on key results and their interpretation.
6. Conclusions
It is not found.
Recommendations: Include a brief and direct conclusion that responds to the question or hypothesis raised in the introduction, without extrapolating or speculating.
7. Acknowledgements and Conflicts of Interest
Absent.
Recommendations Add this section including funding sources, collaborators, and a conflict-of-interest statement.
8. Figures and tables
Supplementary figures and tables are well prepared, but it is recommended: a) ensure that all are referenced in the text; b) verify its clarity and correspondence with the results presented.
9. Strengths
Relevant and current topic. Adequate statistical analysis. Use of figures and tables to support the results.
10. Weaknesses
Lack of clarity in the summary and introduction. Insufficiently developed materials and methods.
11. Additional recommendations
Review the Plants journal guidelines and adjust the manuscript accordingly. Request a critical review before submitting the work again.
12. Final evaluation
The manuscript has potential, but requires adjustments in structure, conciseness, and compliance with journal guidelines. In its current state, it is not suitable for publication. With the recommended improvements, it could be considered for future review.
Author Response
Reviewer 1
Below, you will find the observations and comments made during the review and evaluation of your manuscript, which should be considered for the updating of your manuscript.
Regards, reviewer
Overall evaluation
The submitted manuscript is extensive and addresses a relevant topic. However, it does not fully comply with the guidelines provided by Plants magazine. Here are the observations, strengths, weaknesses, and recommendations to improve the document.
- Summary
The problem is not clearly defined. Absent methodology. Present results. Unidentified conclusion.
Recommendations: Write the abstract in a concise manner including problems, methods, key results, and conclusions. Reduce the length to fit the required format.
Response: Esteemed reviewer, we have meticulously revised the abstract in alignment with your generous suggestion. The abstract has been condensed to comply with journal requirements, covering issues, methods, principal findings, and conclusions.
- Introduction
The context is not clearly identified. The problem is absent. The theoretical or practical justification is not mentioned. There is the experimental design.
Recommendations: Including relevant information. Include a clear context, highlight the problem, and justify the research theoretically and practically.
Response: Dear reviewer, we modified the introduction part by omitting extraneous lines, incorporating statements that emphasize the problem, and providing both theoretical and applied justifications for the research, following your insightful suggestion.
- Materials and Methods
The experimental design is not described. Variables and replicability are not identified. It is not mentioned whether the research requires the endorsement of the Ethics Committee.
Recommendations: Prepare this section including details of the experimental design, variables, controls, replications, and ethical approvals, when applicable. It is important to include information on legal and ethical permits related to the use of plant species, such as those required by national or international regulations (e.g., CITES, Nagoya Protocol). In addition, if the collection was carried out in protected areas or with associated traditional knowledge, permit numbers and compliance with local regulations must be indicated.
Response: Dear reviewer, an explanatory remark on replicability has been included in the methods section, and endorsement from an ethics committee is not applicable in this context. Moreover, the transcriptome raw data have been submitted to the SRA database of the NCBI (PRJNA1208947). We have also added the statement to the corresponding section.
- Results
Present statistical results. The use of tables and figures to summarise the findings is observed.
Recommendations: Check if the results are presented in order of importance and if the figures/tables complement the text efficiently.
Response: Respected reviewer, we have verified that the tables, figures, and results in the results section are displayed in the correct order. Furthermore, the texts are adequately incorporated in every portion.
- Discussion
A synthesis of the results is found. Adequate comparison with other studies. An explanation of the mechanisms is included. Limitations are mentioned. Future hypotheses are proposed.
Recommendations: Reduce the length by eliminating redundancies and better organise the information. Focus on key results and their interpretation.
Response: Dear Reviewer, we have revised the Discussion section in accordance with your recommendations. We have shortened the discussion part by eliminating redundant information and succinctly presenting the relevant results and interpretations.
- Conclusions
It is not found.
Recommendations: Include a brief and direct conclusion that responds to the question or hypothesis raised in the introduction, without extrapolating or speculating.
Response: Esteemed reviewer, A concise conclusion section has been incorporated, addressing the problem or idea presented in the introduction, without extrapolation or speculation following your kind recommendation.
- Acknowledgements and Conflicts of Interest
Absent.
Recommendations Add this section including funding sources, collaborators, and a conflict-of-interest statement.
Response: Dear reviewer, the conflict of interest and funding section is included in the article.
- Figures and tables
Supplementary figures and tables are well prepared, but it is recommended: a) ensure that all are referenced in the text; b) verify its clarity and correspondence with the results presented.
Response: Esteemed reviewer, Thank you for your supportive statement. We confirm that all data (tables, figures, and extra information) are clearly provided in the results section. Furthermore, we incorporated supplementary figures to improve the integrity of our article.
- Strengths
Relevant and current topic. Adequate statistical analysis. Use of figures and tables to support the results.
Response: Esteemed reviewer, Thank you for your supportive statement.
- Weaknesses
Lack of clarity in the summary and introduction. Insufficiently developed materials and methods.
Response: Regarded reviewer, We have improved the introductory section for clarity and integrated a comprehensive approach in the methodology section.
- Additional recommendations
Review the Plants journal guidelines and adjust the manuscript accordingly. Request a critical review before submitting the work again.
Response: Dear reviewer, we have adhered to the journal's criteria, and the paper complies with all requirements.
- Final evaluation
The manuscript has potential, but requires adjustments in structure, conciseness, and compliance with journal guidelines. In its current state, it is not suitable for publication. With the recommended improvements, it could be considered for future review.
Response: Esteemed reviewer, we have meticulously rewritten the paper in accordance with the reviewers' suggestions, incorporating an additional figure and a conclusion section, along with the previously omitted explanatory lines. We deeply value your constructive recommendation to enhance our manuscript.
Reviewer 2 Report
Comments and Suggestions for Authors
The manuscript by Ke et al. is devoted to analysis of role of anthocyanin and chlorophyll metabolic pathways in the floral sepals color formation in different hydrangea cultivars. The work seems to be interesting and have practical perspective. However, there are comments and questions, which should be discussed.
1. P. 3, lines 112-113: “To assess the sepal color, we employed the CIELAB technique to specify multiple leaf color indices (L*, a*, b*) and analyze pigment concentrations”. What is this CIELAB technique? It is not described in Results and/or Materials and Methods. This technique should be described in more detail.
2. Authors analyzed content of chlorophylls in color sepals on based of SPAD. Is it correct? SPAD are widely used for leaves; however, can it be used to estimate concentration of chlorophylls in color sepals? This methodical point should be considered.
3. Figure 2A: There are three points per each variant (GB1, GB2, and GB3; DB1, DB2, and DB3; LB1, LB2, and LB3). What do these points show? In the next section, similar points are described as “biological replicates” (P. 5, lines 178-179). If these points are biological replicates in Figure 2A, too, it should be described. Additionally, were three biological replicates statistically enough?
4. Figure 2C: Only delphinidin is clearly shown in this figure; other anthocyanins are weakly shown. It is not suitable for potential readers. I suppose different histograms should be used for different anthocyanins.
5. Figure 2D: What is parameter showed in heat map? What does color scale show? It should be clarified.
6. Captions of Figures 2 and 3: “(D) Heat map, (E) and k-means cluster analysis…” seems to be confused. Maybe “(D) Heat map, and (E) k-means cluster analysis…”?
Author Response
The manuscript by Ke et al. is devoted to analysis of role of anthocyanin and chlorophyll metabolic pathways in the floral sepals color formation in different hydrangea cultivars. The work seems to be interesting and have practical perspective. However, there are comments and questions, which should be discussed.
Question/suggestion: 1. P. 3, lines 112-113: “To assess the sepal color, we employed the CIELAB technique to specify multiple leaf color indices (L*, a*, b*) and analyze pigment concentrations”. What is this CIELAB technique? It is not described in Results and/or Materials and Methods. This technique should be described in more detail.
Response: Esteemed reviewer, The CIELAB nonlinear adjustment of the RGB color space equates the actual distance between two colors to their Euclidean distance for distances under 10 units. CIELAB frequently produces superior results in color image processing techniques. We have also included references in the methodology section.
Question/suggestion: Authors analyzed content of chlorophylls in color sepals on based of SPAD. Is it correct? SPAD are widely used for leaves; however, can it be used to estimate concentration of chlorophylls in color sepals? This methodical point should be considered.
Response: Esteemed reviewer, yes, we employ SPAD to assess the chlorophyll levels in colored sepals. Figure 1 illustrates that hydrangea is characterized by its sepals rather than true petals, which enables color variation. Furthermore, its configuration resembles that of leaves. Consequently, we employed SPAD to quantify chlorophyll content, yielding successful measurements. We utilize numerous replicates to substantiate the authentication of our outcomes.
Furthermore, we have replicated and reanalyzed the chlorophyll content of hydrangea sepals using a spectrophotometer, yielding results that corroborate the SPAD values. In accordance with your recommendation, we substituted Figure 1E (SPAD values) with total chlorophyll content.
Question/suggestion: Figure 2A: There are three points per each variant (GB1, GB2, and GB3; DB1, DB2, and DB3; LB1, LB2, and LB3). What do these points show? In the next section, similar points are described as “biological replicates” (P. 5, lines 178-179). If these points are biological replicates in Figure 2A, too, it should be described. Additionally, were three biological replicates statistically enough?
Response: Esteemed reviewer, GB, DB, and LB represent cultivar names, whereas GB1-3 indicate replicates. Indeed, esteemed reviewer, three replicates are statistically sufficient and have been utilized in numerous investigations. We have illustrated the replicates in the legends of Figure 2 as per your request.
Question/suggestion: Figure 2C: Only delphinidin is clearly shown in this figure; other anthocyanins are weakly shown. It is not suitable for potential readers. I suppose different histograms should be used for different anthocyanins.
Response: Dear reviewer, delphinidin was the predominant compound identified in the cultivars; nonetheless, in accordance with your recommendation, we have incorporated histogram statistics in the supplemental data (Supplementary figure S1) for the benefit of the readers.
Question/suggestion: Figure 2D: What is parameter showed in heat map? What does color scale show? It should be clarified.
Response: Dear reviewer, the heatmap was generated using normalized data, with a color scale indicating the high and low expression levels of differentially expressed genes (DEGs) across three hydrangea cultivars. We have incorporated the statement in the figure caption as per your suggestion.
Question/suggestion: Captions of Figures 2 and 3: “(D) Heat map, (E) and k-means cluster analysis…” seems to be confused. Maybe “(D) Heat map, and (E) k-means cluster analysis…”?
Response: Dear reviewer, we have substituted the ambiguous statement with your proposed statement. We sincerely value your astute observations and significant contributions to enhancing the manuscript's quality.
Round 2
Reviewer 2 Report
Comments and Suggestions for Authors
Authors completely considered my comments and improved the manuscript. I suppose that this interesting work can be accepted.